:ۥ: PLOS | ONE

# Do protection gradients explain patterns in herbivore densities? An example with ungulates in Zambia's Luangwa Valley

Elias Rosenblatt[1,2¤]*, Scott Creel[1,2], Paul Schuette[1,3], Matthew S. Becker[1,2], David Christianson[1,4], Egil Dröge[5], Thandiwe Mweetwa[1], Henry Mwape[1], Johnathan Merkle[1], Jassiel M'soka[1], Jones Masonde[6], Twakundine Simpamba[6]

**1** Zambian Carnivore Programme, Mfuwe, Eastern Province, Zambia, **2** Department of Ecology, Montana State University, Bozeman, Montana, United States of America, **3** Alaska Center for Conservation Science, University of Alaska Anchorage, Anchorage, Alaska, United States of America, **4** School of Natural Resources and the Environment, University of Arizona, Tucson, Arizona, United States of America, **5** Wildlife Conservation Research Unit, University of Oxford, Tubney, United Kingdom, **6** Department of National Parks and Wildlife, Chilanga, Zambia

¤ Current address: Rubenstein School of Environment and Natural Resources, University of Vermont, Burlington, Vermont, United States of America
* erosenbl@uvm.edu

**Data Availability Statement:** All relevant data are within the manuscript and its Supporting Information files.

## Abstract

Ungulate populations face declines across the globe, and populations are commonly conserved by using protected areas. However, assessing the effectiveness of protected areas in conserving ungulate populations has remained difficult. Using herd size data from four years of line transect surveys and distance sampling models, we modeled population densities of four important herbivore species across a gradient of protection on the edge of Zambia's South Luangwa National Park (SLNP) while accounting for the role of various ecological and anthropogenic variables. Our goal was to test whether protection was responsible for density dynamics in this protection gradient, and whether a hunting moratorium impacted herbivore densities during the studies. For all four species, we estimated lower densities in partially protected buffer areas adjacent to SLNP (ranging from 4.5-fold to 13.2-fold lower) compared to protected parklands. Density trends through the study period were species-specific, with some species increasing, decreasing, or remaining stable in all or some regions of the protection gradient. Surprisingly, when controlling for other covariates, we found that these observed differences were not always detectably related to the level of protection or year. Our findings highlight the importance of accounting for variables beyond strata of interest in evaluating the effectiveness of a protected area. This study highlights the importance of comprehensively modeling ungulate population density across protection gradients, identifies lands within an important protection gradient for targeted conservation and monitoring, documents prey depletion and expands our understanding on the drivers in a critical buffer area in Zambia.

**Funding:** This research was funded by: 1. WorldWide Fund for Nature–Netherlands, https://www.wnf.nl/, Grantees: Matt Becker; 2. National Science Foundation Animal Behavior Program under IOS-1145749, Website: https://www.nsf.gov/funding/aboutfunding.jsp, Grantees: Scott Creel, Matt Becker, Dave Christianson, Paul Schuette; 3. National Geographic's Big Cats Initiative, Website: https://www.nationalgeographic.org/projects/big-cats-initiative/, Grantees: Matt Becker; 4. Painted Dog Conservation Inc, Website: http://www.painteddogconservation.iinet.net.au/, Grantees: Matt Becker. The funders had no role in study design, data collection and analysis, decision to publish, or preparation of the manuscript.

**Competing interests:** The authors have declared that no competing interests exist.

# 1 Introduction

In sub-Saharan Africa, ungulates have ecological and economic value through their top-down effects on plant communities and their bottom-up effects as prey for large carnivores [1]. Despite their importance, many protected ungulate populations have recently declined at a rate comparable to populations with less protection [2,3] and face rapid human encroachment [4]. Protected areas (PAs) are an important tool to protect wildlife from human activities and reduce human-wildlife conflict [5]. Strictly protected areas with no permanent settlements and no consumptive use are often bordered by areas with some lower level of protection, creating a gradient to buffer edge effects [6] and source-sink dynamics [7]. However, it remains unclear how effective these protection gradients are in protecting ungulate species of ecological importance or conservation concern [5].

Despite the intuitive benefits of PAs, assessing the effectiveness of PAs in protecting ungulate populations is difficult. First, to provide a valid test of the effect of protection gradients, studies must control for ecological differences between PAs and adjacent buffer zones that could affect ungulate density and distribution. Protected areas are not randomly distributed [8] but instead are generally placed where wildlife densities are high, while buffer zones are often designated in areas with lower wildlife densities. Thus, differences in animal density between PAs and buffer zones can exist due to natural ecological differences between locations unrelated to the effectiveness of their protection. Second, ungulates cannot be surveyed with perfect detection in most habitats [9]. Methods exist to account for imperfect detection when estimating animal densities [10,11], but many studies of protection gradients assume perfect detection or use an index to convert counts that rely on untested assumptions about detection probability. Finally, even when methods that account for detection are used, a common approach is to model the density of groups with distance sampling [10] and then convert group density to individual density using a mean group size [10,12]. This conversion uses either mean group size across all observations or across focal categorical variables, such as vegetation types [13]. Because ungulate group size is typically influenced by the same variables that affect the distribution of herds, this approach may not be entirely accurate. Important broad studies have aided our understanding of the effectiveness of PAs to conserve ungulates [14–17], but their inferences have been constrained by one or more of these limitations.

Zambia contains several PAs important for regional conservation of ungulates and the large carnivores that depend on ungulate prey [18], and most of these PAs face human encroachment that is approaching rapidly or already has reached the PA itself [19]. Most Zambian PAs are buffered by Game Management Areas (GMAs) that allow some human settlement and support consumptive uses of wildlife and resources, such as legal trophy hunting [20]. Rapid human population growth in GMAs has brought increased pressures of illegal bushmeat harvest and habitat conversion, challenging the effectiveness of Zambia's GMAs as buffers for adjacent PAs [21]. South Luangwa National Park (SLNP) is Zambia's premier PA for wildlife-based photo-tourism and conserves regionally-important populations of several threatened and endangered species yet faces rapid human pressure from adjacent GMAs [19]. Recent studies of large carnivore demography and dynamics [20,22–25] and studies of bushmeat poaching patterns [21,26,27] suggest that the depletion of ungulate populations from bushmeat poaching may be affecting the ecological integrity of this protection gradient. Aerial surveys also suggest overall ungulate declines in the Luangwa Valley and lower ungulate density in GMAs compared to PAs [21], but these studies are generally inconclusive due to low precision of density estimates and no correction for variability in detection. Finally, it has been suggested that a temporary moratorium in all trophy hunting from 2013 to 2014 allowed increased poaching activity in the absence of the primary wildlife-based tourism activity in GMAs [28]. Despite this concern and reported increased poaching in

communities in GMAs adjacent to SLNP [29], there has not been sufficient data to test the relationship between the hunting moratorium and herbivore population trends. Therefore, to better inform management actions in this important PA, there is a clear need for unbiased and precise estimates of ungulate abundance in SLNP and adjacent GMAs, for tests of the relationship between ungulate density and the level of protection, and for tests of trends in density over time and across management actions, while addressing the difficulties introduced above.

Here we use data from repeated, stratified line transect sampling to fit distance sampling and group size models to estimate population densities of impala (*Aepyceros melampus*), puku (*Kobus vardonii*), plains zebra (*Equus quagga*), and warthog (*Phacochoerus africanus*) across the South Luangwa Protection Gradient from 2012–2015. These species are abundant in the area, important prey for carnivore species of concern, primary targets for illegal bushmeat trade [30] and are expected to be negatively impacted by the absence of trophy hunting during the 2013–2014 moratorium. For each species we model both group density and group size as functions of top-down, bottom-up, abiotic, and anthropogenic covariates, and then estimate differences in population density across space and time while controlling for these effects. Our approach addresses the above difficulties in assessing the efficacy of protected areas as it accounts for imperfect detection, robustly integrates ecological and anthropogenic covariates that impact both the distribution and size of ungulate groups, compares ungulate densities within a PA and GMA that are similar ecologically but differ in human usage, and assess population trends with the temporary cessation of trophy hunting activities. The results improve our understanding of the efficacy of protection gradients in buffering the impacts of human encroachment and bushmeat poaching and provide baseline estimates of density for ecologically important ungulate populations that face increasing anthropogenic pressures.

## 2. Materials and methods

### 2.1 Study area

Our study area (hereafter the South Luangwa Protection Gradient, or SLPG) was within a 1 200 km$^2$ complex of grasslands, scrublands, and forests situated along the perennial Luangwa River, the primary eastern boundary of SLNP and the adjacent Lupande GMA (hereafter the GMA; Fig 1) [31,32]. The Luangwa River is only a seasonal barrier for wildlife and attracts the highest densities of wildlife in the area as it is the largest and most reliable perennial water source in the region, particularly during the dry season (May-November).

The SLPG includes three regions of interest for this study. West of the Luangwa River are fully protected SLNP lands (818 km$^2$), where the primary human activity is photographic safari tourism and Park management activities. East of the Luangwa River there is a section of SLNP (151 km$^2$) that is also popular for photographic tourism but is thought to be exposed to more illegal bushmeat poaching due to open borders and a public road that bisects the area. While this is a small, unsurfaced dirt track, it supports considerable local foot and bicycle traffic, and some vehicular traffic. Finally, the GMA lands (231 km$^2$), also east of the Luangwa River, contain a growing human population (annual growth of 3.8%) [33] that has raised an array of conservation concerns (see Section 1). Trophy hunting operates in leased, unfenced concessions in the GMA (save for a moratorium from 2013–2014) [20], targeting an array of wildlife species including those studied here [31]. In short, our study area encompasses areas of high conservation importance across a protection gradient with associated variation in human influence and other potentially limiting factors.

### 2.2 Study design

**2.2.1 Survey design.** We used line transects and distance sampling [10] to estimate herd density (herds/km$^2$) and herd size (individuals/herd) for ungulates across the SLPG. We

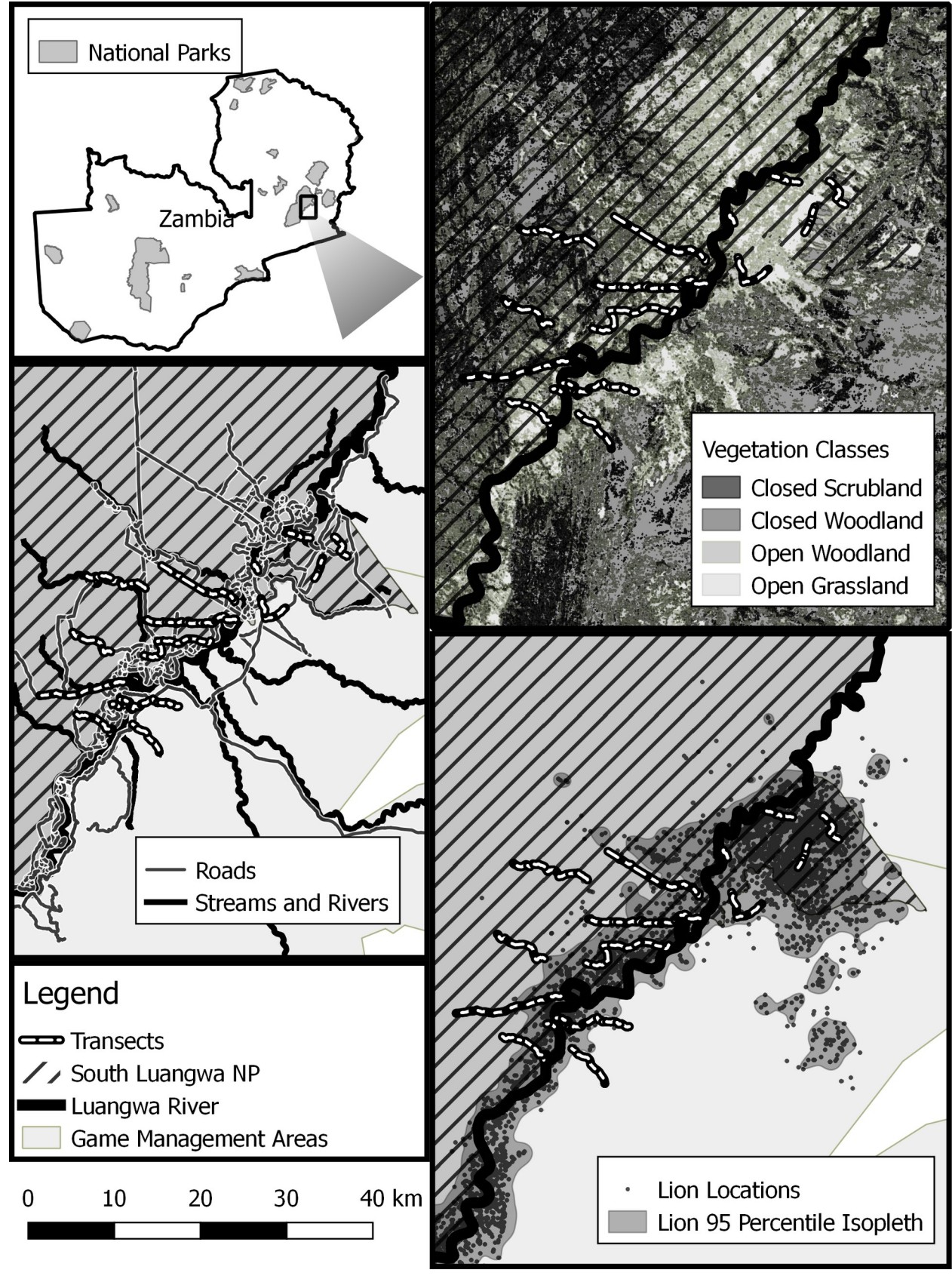

**Fig 1. Our intensive study area (1 200 km²) on the eastern boundary of South Luangwa National Park (SLNP; S13.07958 E31.77407), Zambia, faceted by key spatial covariates.** Line transects (n = 15) were stratified across bottom-up, top-down, anthropogenic, and abiotic covariates, and each transect was comprised of multiple segments (n = 97; not shown). Maps are faceted to illustrate key spatial covariates. Vegetation classes were defined from Landsat 7 Global Land Survey 2010 imagery. Lion locations reflect pride and coalition locations centering around perennial water sources during the dry season, despite intensive and constant lion monitoring across the 1200 km study area. Group observations and covariates were recorded by segment (see Section 2.2.2).

surveyed 15 transects by vehicle across our study area (120.5 km in total), with sufficient spacing between transects to minimize the risk of double-counting groups [10] (Fig 1). We established most transects in a generally east-west orientation (i.e., perpendicular to the Luangwa River) to sample across the range of environmental and anthropogenic variables of interest. Topography and vegetation impassable by vehicle forced us to follow roads for sections of transects, which we accounted for in our modeling of detection (See Section 2.2.2). Each transect was split into segments defined by observed changes in vegetation cover type or a maximum length of 2km, resulting in a total of 97 segments across the three regions of interest in the SLPG (West of the Luangwa and National Park, n = 62; East of the Luangwa and National Park, n = 15; GMA, n = 20). These segments allowed us to estimate the role of covariates on a finer scale than for entire transects (e.g. distance to the Luangwa River; See Section 2.2.2). We were not able to achieve balanced sampling across the SLPG due to human settlement and agriculture activity in the GMA. GMA transects were primarily on primitive roads or off-road, to minimize any sensitivity of animals to vehicle noise. As illegal harvests of wildlife are usually done on foot, these species did not appear to be sensitive to our driven transects. We recorded herbivore group observations and covariate values on the segment level as required by our analysis described below.

We conducted 10 surveys of all 97 segments during daylight hours (0900–1700) at the beginning (June), middle (August), and end (October) of the dry season (May-November) from 2012–2015 (n = 970 segment-surveys). June 2012 and August 2015 were not surveyed due to logistic constraints. We could not conduct any surveys during the rainy season (December-April) as much of the study area is inaccessible. During each survey, the vehicle was driven at a maximum speed of 15 km/h, with two observers seated on the rooftop scanning for animal herds. When a herd or single animal was seen, the observers recorded the species composition and size of the herd, and the distance (aided by a laser rangefinder) and azimuth to the herd following standard distance sampling protocol [10]. We reduced our risk of double-counting animal herds during each survey by surveying all segments over a period of 3–4 days.

**2.2.2 Factors affecting group density and size.** We considered a suite of bottom-up, top-down, anthropogenic, and abiotic variables that could affect herd density and herd size of the focal species on the segment level (Table 1). We measured ten covariates that characterize bottom-up effects of vegetation composition and structure on herd density and herd size. While surveying segments we recorded the presence or absence of green grass and of grassy lagoon patches, and the predominant grass height category (short = <10cm, intermediate: 10-100cm, long: >100cm). We suspected that vegetation composition would influence both herd density and herd size, and our ability to detect herds. To model detection, we assigned each segment to one of six vegetation structure categories, as well as one of three simplified vegetation categories (Table 1): because transects were split into segments at irregular intervals that were defined by changes in vegetation structure, each segment was relatively homogenous in structure. We considered these two structure covariates separately during our analyses as they were two descriptions of the same underlying variable. To model group density and herd size, we classified vegetation composition and heterogeneity within a 240m-wide buffer around each segment (hereafter buffer) to characterize vegetation beyond the view of the survey team. Using Landsat 7 Global Land Survey 2010 imagery we partitioned pixels into 10 vegetation

**Table 1. A summary of all covariates that were thought to impact herbivore density, classified as Bottom-up (B), Top-down (T), Anthropogenic (A), or Abiotic (AB).** Covariates were considered in distance sampling models to impact group super-population ($\lambda$), availability for sampling ($\varphi$), or detection ($p$). Covariates were (Y) or were not (N) considered in models predicting group size.

| Covariate | Factor Class | Categorical Levels | Continuous Range (Mean) | Distance Sampling | Group Size |
|---|---|---|---|---|---|
| Edge Density (km/km2) | B | - | 3.5–55.6 (24.4) | $\lambda$ | Y |
| Grassy Lagoon | B | Present, Absent | - | $\varphi$ | Y |
| Percent Closed Scrub | B | - | 0–38.0 (2.3) | $\lambda$ | Y |
| Percent Closed Woodland | B | - | 0–90.8 (23.4) | $\lambda$ | Y |
| Percent Open Woodland | B | - | 0.5–96.3 (37.7) | $\lambda$ | Y |
| Percent Open Grassland | B | - | 0–99.5 (36.7) | $\lambda$ | Y |
| Grass Height | B | Short, Intermediate, Tall | - | $\varphi$ | Y |
| Grass Color | B | Green, Brown | - | $\varphi$ | Y |
| Vegetation/Density Class | B | Closed Scrub, Closed Woodland, Open Scrub, Open Woodland, Open Grassland Flooded, Open Grassland Not Flooded | - | $p$ | N |
| Simplified Vegetation Class | B | Scrub, Woodland, Grassland | - | $P$ | N |
| Lion UD | T | - | 6.0–121.1 (49.0) | $\lambda$ | Y |
| Distance to Roads (km) | A | - | 0–2.4 (0.3) | $\lambda$ | Y |
| Side of River | A | East, West | - | $\lambda$ | Y |
| Area | A | GMA, Park | - | $\lambda$ | Y |
| Segment Path Type | A | Off-road, Seasonal Track, Gravel Road | - | $p$ | N |
| Distance to Luangwa River (km) | AB | - | 0.2–14.7 (4.1) | $\lambda$ | Y |
| Distance to Seasonal Stream (km) | AB | - | 0–9.4 (3.4) | $\lambda$ | Y |
| Water | AB | Present, Absent | - | $\varphi$ | Y |
| Burn | AB | Present, Absent | - | $\varphi$ | Y |
| Year | AB | 2012, 2013, 2014, 2015 | - | $\varphi$ | Y |
| Dry Season Stage | AB | Early, Middle, Late | - | $\varphi$ | Y |

classes of differing vegetation densities using the clara() function in the *cluster* package [34] in R [35]. We then combined vegetation classes into four cover types based on vegetation density and validated these cover type classifications during our line transect surveys. These cover types included closed scrubland (right-skewed, and therefore log-transformed), closed woodland, open woodland, and open grassland. We estimated the proportion of buffer composed of each cover type for each segment using the *sp* package in R [36,37] and estimated the density of edges between cover types within each buffer (km edge/km$^2$) using the perimeter tool in QGIS 2.0.1 (www.qgis.org).

Differences in predation risk could affect herd density or herd size across the SLPG, thus we quantified predation risk across our study area by measuring the utilization of the area by the lion (*Panthera leo*) population (the most abundant large carnivore within the study area) [20]. Intensive lion population monitoring was ongoing during this study, with prides and coalitions monitored across the 1200 km study area detailed in this manuscript. We fit a single kernel utilization distribution to 7 785 lion locations collected over five years (2008–2012) from lions equipped with GPS radiocollars and direct observation of 18 lion prides and 14 male coalitions [20]. Though this spatial information overlaps only one year of this study, ongoing lion studies indicate little change in lion distribution [25]. We used *sp*, *rgdal* [38], and *plyr* [39] packages in R to estimate the distribution of daily distance moved from six lionesses

equipped with GPS radiocollars (range: 0m - 17 300m) and used the 90th percentile (6 974m) as a smoothing parameter for the utilization distribution (UD) [40]. We fit the UD using the bivariate normal kernel function in the *adehabitatHR* package [41], with an output grid of 300m x 300m. We converted the UD to a raster using the *raster* [42] and *sp* packages and extracted and standardized lion UD values for each segment's midpoint, thereby quantifying the risk of herbivores encountering the dominant large carnivore species in each segment.

The impact of anthropogenic activities was characterized for each segment by classifying whether it lay within the PA or partially-protected GMA (hereafter protection), on the east or west side of the Luangwa River (hereafter side), and the distance from the segment midpoint to the nearest road (also right-skewed, and log-transformed). We could not test for an interaction between protection and side as there are no GMA areas within the study area on the western side of the Luangwa River. We also could not include distance to park boundary in our analysis as this boundary is mostly defined by the Luangwa River. Despite the Luangwa River defining a boundary in the SLPG, any effect of this boundary in our study is confounded by the abiotic influence of the Luangwa River as the SLPG's perennial water source. Therefore, we chose to treat the Luangwa River as an abiotic variable, and not an anthropogenic variable (see below). Finally, as some segments followed roads, we classified each segment's path type to account for any biases in detection for road-based segments (Table 1) [43].

We measured six abiotic covariates to characterize the environmental conditions potentially influencing herbivore density and distribution in the Luangwa Valley (Table 1). Availability of water is a limiting factor for wildlife in the Luangwa Valley, with water sources diminishing and disappearing with the progression of the dry season. Therefore, we measured distances from each segment's midpoint to the perennial Luangwa River and the nearest seasonal stream and recorded the dry season stage and whether standing water was present during surveys of each segment. We also recorded the year of each survey to determine annual trends in group density and size. Finally, we recorded whether there was evidence of a fire that had burned through the segment for each survey, as post-fire "green flush" provides herbivores with high-quality forage and potentially reduces predation risk [44].

## 2.3 Analytical methods

**2.3.1 Herd density analysis.** We used a multinomial generalized distance sampling model to estimate the 'super-population' of herds ($\lambda$) for each species within each segment, while accounting for imperfect detection ($p$) and varying availability for detection at the time of the survey ($\varphi$)[45]. We used the gdistsamp() function to fit candidate models for each species in the *unmarked* package [45] in R. We truncated herd sighting data for each species to exclude outlier distances that would compromise estimation of detection probability.

To focus on accurate estimation of the parameter of interest (group density), we evaluated a set of models in three steps using Akaike's Information Criteria (AIC; Table 2). We first identified the best-supported model for detection across all combinations of covariates thought to influence detection using hazard, half-normal, and uniform detection functions, while estimating only a mean for availability and super-population (Tables 1 and 2). Using the best-supported detection model and a uniform super-population model, we next identified the best supported model for availability across all combinations of covariates thought to influence availability (Tables 1 and 2). In the third step, while modeling detection and availability using their best-supported models, we used AIC model selection to identify the best supported models ($\leq$ 1 AIC) for each of the four types of effect on the super-population of herds: bottom-up, top-down, anthropogenic, and abiotic (Table 2). Finally, we took the top model for each of these four types and used AIC scores to identify the final density models ($\leq$ 1 AIC), selecting

**Table 2. Full models of detection (p), availability (φ), and group super-population (λ) to illustrate the model refinement process.** We used AIC model selection to evaluate these models and their subsets for each species and used the best supported models for each parameter to build the final model set. For continuous covariates we considered linear, log, and second-order polynomial relationships. For super-population, we identified the best performing model for each covariate type, and then created a final candidate model list using all combinations of those parametrizations. We split bottom-up abundance model selection into vegetation availability (proportion of vegetation classifications around segments) and edge density model sets to reduce computation time.

| Step | Parameter | Detection | Availability | Super-population |
|------|-----------|-----------|--------------|------------------|
| 1 | $p$ | ~ Vegetation/Density Class + Path Type **or** ~ Vegetation Class + Path Type | ~1 | ~1 |
| 2 | $φ$ | $p(Top)$ | ~Dry Season Stage + Year + Grass Height + Grass Color + Burn + Grassy Lagoon + Water | ~1 |
| 3 | λ(Abiotic) | $p(Top)$ | $φ(Top)$ | ~ Distance to Luangwa River + Distance to Seasonal Stream |
|  | λ(Bottom-Up: Edge Density) | $p(Top)$ | $φ(Top)$ | ~ Edge Density |
|  | λ(Top-Down) | $p(Top)$ | $φ(Top)$ | ~ Lion UD |
|  | λ(Bottom-Up: Vegetation Availability) | $p(Top)$ | $φ(Top)$ | ~ log(% Closed Scrub) + % Closed Woodland + % Open Woodland + % Open Grassland |
|  | λ(Anthropogenic) | $p(Top)$ | $φ(Top)$ | ~ log(Distance to Roads) + Area + Side of River |
|  | **λ(Final)** | **$p(Top)$** | **$φ(Top)$** | **~ λ(Abiotic-*Top*) + λ(Edge Density-*Top*) + λ(Top-Down-*Top*) + λ(Vegetation Availability-*Top*) + λ(Anthropogenic-*Top*)** |

from a set with all additive combinations of effects in the top models for each type (Table 2). This multi-stage approach to limit the number of models compared was necessary to reduce computation time. We used deviance goodness-of-fit tests for each species' top herd density model(s) to examine model fit. We used these final models and model averaging (when more than one model received comparable support from the data) to estimate herd super-population ($λ$) and availability ($φ$) using the *predict()* function, and derive herd density estimates ($\hat{D}$) as the product of these two parameter estimates.

**2.3.2 Herd size analysis.** We fit zero-truncated Poisson (ZTP) regression models using the *vglm()* function in the VGAM package [46,47] in R to estimate segment-specific mean herd sizes for the four focal species while controlling for the effects of covariates described above (Table 1). First, we used a ZTP model to test whether herd size was affected by distance from the transect, as is expected if herd size affects detection, and only used herd size observations at 0m from the transect if this there was evidence of this effect (p < 0.15) [10]. Next, we dropped covariates with badly imbalanced sampling or that were highly correlated with other covariates (Pearson's r > 0.6) from consideration. We then used reverse step-wise likelihood-ratio (LR) tests to select a herd size model for each species, and then confirmed this model selection using forward step-wise LR tests. We evaluated model fit by fitting a linear regression of Pearson's residuals on predicted herd sizes and assumed adequate fit if the estimated intercept and slope were not detectably different from 0 [46]. Using the final ZTP model and the *predict()* function we estimated differences in herd size across covariate ranges and variation in herd size while accounting for the non-random distribution and correlation of covariates across the study area. To avoid extrapolation, we only estimated mean herd size for segments with covariate values within the range documented during herd observations.

**2.3.3 Population density analysis.** To estimates population densities for each species, we multiplied mean herd density by estimates of mean herd size for each segment and used non-parametric bootstrapping to estimate mean population density and its variance across SLPG regions and years. We also predicted herd density and herd size with all covariates other than

protection, side, and year fixed at their mean value, estimated mean population density and variance across those strata of interest, and calculated differences across SLPG regions and across years. In summary, our comprehensive approach allows a test for the effect of protection and year on population density that does not ignore the ecological and abiotic differences across the protection gradient.

# 3. Results

## 3.1 Herd density and herd size

During the 10 surveys of 97 segments, we detected 890 impala herds, 478 puku herds, 175 zebra herds, and 169 warthog herds. After truncating datasets to maintain suitable detection probabilities (400m for puku, 300m for the other three species), our final sample sizes for our herd density analysis were 836 impala herds, 452 puku herds, 155 zebra herds, and 163 warthog herds (S1 Fig). Distance had a positive association with impala and puku herd size (p<0.0001 and p = 0.005, respectively), but not with zebra and warthog herd size (p = 0.43 and p = 1, respectively). These results indicated that large impala and puku herds were more detectable at large distances, so to avoid bias we only used herds that were directly on the transect to estimate herd size for these two species. Our final sample for our herd size analysis included 122 impala herds (mean = 7.1 individuals, range: 1–75), 56 puku herds (mean = 6.6 individuals, range: 1–111), 155 zebra herds (mean = 5.1 individuals, range: 1–24), and 163 warthog herds (mean = 2.6 individuals, range: 1–9).

The best supported herd density model(s) for each parameter varied between species (Table 3), and coefficient estimates varied in magnitude and sign between models (Table 4). Despite this variation two variables of primary interest—side of Luangwa River and protection status—were included in top herd density models for all species. Herd density was higher in the National Park for all species. Segments west of the Luangwa River were estimated to have higher impala and zebra herd densities, whereas puku and warthog densities were lower or not detectibly different than in segments east of the Luangwa River, respectively. Survey year influenced availability for zebra and warthog herd density, with herd density detectably higher for zebra in 2015, but detectably lower for warthog in 2015.

Like herd density models, herd size models and coefficient estimates varied between species (Table 5). Segments on the western side of the Luangwa River were associated with smaller herds for impala (21% smaller) and puku (16% smaller), while neither side of the Luangwa River nor protection status were detectably correlated with zebra or warthog herd sizes. Impala and puku herds were smallest in 2012, with detectably larger herds during the remainder of the study period (except 2014 for puku).

## 3.2 Population density

Impala had the highest population density, with an average density of 27.69 animals/km$^2$, and showed great spatiotemporal variation in density (range among segments: 0.24–601.18 animals/km$^2$). Puku also occurred at variable densities, an average density of 8.11 animals/km$^2$ (range: 0.003–172.60 animals/km$^2$). Zebra and warthog occurred at lower and less variable densities, with average densities of 2.41 animals/km$^2$ (range: 0.03–28.09 animals/km$^2$) and 1.76 animals/km$^2$ (range: 0.05–13.82 animals/km$^2$), respectively.

**3.2.1 Protection effects.** Herbivore densities varied widely across the SLPG, and the highest densities for all species were within the fully-protected SLNP (Fig 2). Within SLNP, density varied between portions of the park that were west and east of Luangwa River. Impala and zebra densities did not detectably differ within these two areas of the national park, while both puku and warthog occurred at their highest densities in eastern parklands. Thus, there were no

**Table 3. Model selection results for distance sampling models estimating group density for the focal herbivore species.** Model-averaged predictions were made using models within 1 delta AIC score of the top model; top models were used if there were no closely-competing models. Continuous covariates may appear in models as a linear, logarithmic (log), or 2nd-order polynomial (poly) association with the parameter.

| Model—λ | Model—φ | Model—p | Parameters | delta AIC | AIC Weight |
|---|---|---|---|---|---|
| Impala | | | | | |
| ~poly(distance to seasonal stream)+ poly(Distance to Luangwa River) + poly(edge density)+log(lion UD)+log(distance to road)+area+side of river | ~dry season stage + grass height + burn + lagoon + water | ~Vegetation/ Density | 26 | 0 | 0.72 |
| ~poly(distance to seasonal stream)+ poly(Distance to Luangwa River) + poly(edge density)+log(lion UD)+ log(% Open Grassland)+ log(distance to road)+area+side of river | ~dry season stage + grass height + burn + lagoon + water | ~Vegetation/ Density | 27 | 1.86 | 0.28 |
| Puku | | | | | |
| ~poly(distance to seasonal stream)+ poly(Distance to Luangwa River) + poly(edge density)+poly(lion UD)+log(CW)+log(OG)+poly(log(distance to road))+area +side of river | ~grass height + grass color + burn + lagoon | ~Vegetation+ Path | 27 | 0 | 0.44 |
| ~poly(distance to seasonal stream)+ poly(Distance to Luangwa River) + poly(edge density)+poly(lion UD)+log(CW)+log(OG)+log(CS)+ poly(log(distance to road)) +area+side of river | ~grass height + grass color + burn + lagoon | ~Vegetation+ Path | 28 | 1.07 | 0.26 |
| ~poly(distance to seasonal stream)+ poly(Distance to Luangwa River) + poly(lion UD)+log(CW)+log(OG)+poly(log(distance to road))+area+side of river | ~grass height + grass color + burn + lagoon | ~Vegetation+ Path | 25 | 1.61 | 0.20 |
| Zebra | | | | | |
| ~log(Edge) + log(CW) + log(OW) + area + side of river | ~dry season stage + year + grass color + lagoon | ~Vegetation/ Density + Path | 23 | 0 | 0.13 |
| ~log(CW) + log(OW) + area + side of river | ~dry season stage + year + grass color + lagoon | ~Vegetation/ Density + Path | 22 | 0.37 | 0.10 |
| ~log(distance to road)+log(CW) + log(OW) + area + side of river | ~dry season stage + year + grass color + lagoon | ~Vegetation/ Density + Path | 23 | 0.58 | 0.09 |
| ~log(Edge) + log(CW) + log(OW) + log(distance to road) + area + side of river | ~dry season stage + year + grass color + lagoon | ~Vegetation/ Density + Path | 24 | 0.7 | 0.09 |
| ~log(Edge) + log(CW) + log(OW) + poly(log(distance to road)) + area + side of river | ~dry season stage + year + grass color + lagoon | ~Vegetation/ Density + Path | 25 | 1.43 | 0.06 |
| ~log(CW) + log(OW) + poly(log(distance to road)) + area + side of river | ~dry season stage + year + grass color + lagoon | ~Vegetation/ Density + Path | 24 | 1.56 | 0.06 |
| Warthog | | | | | |
| ~log(Lion_UD)+poly(log(CS))+poly(CW)+poly(OG)+log(distance to road)+Area | ~year + water | ~Vegetation/ Density | 22 | 0 | 0.10 |
| ~log(Lion_UD)+poly(CW)+poly(OW)+poly(OG)+log(distance to road)+Area | ~year + water | ~Vegetation/ Density | 22 | 0.74 | 0.07 |
| ~log(Lion_UD)+poly(CW)+poly(OW)+poly(OG)+Area | ~year + water | ~Vegetation/ Density | 21 | 1.11 | 0.06 |
| ~log(Lion_UD)+poly(log(CS))+poly(CW)+poly(OG)+Area | ~year + water | ~Vegetation/ Density | 21 | 1.14 | 0.06 |
| ~Distance to Seasonal Streams+log(Lion_UD)+ poly(log(CS))+poly(CW)+ poly (OG)+log(distance to road)+Area | ~year + water | ~Vegetation/ Density | 23 | 1.76 | 0.04 |
| ~Lion_UD+poly(log(CS))+poly(CW)+poly(OG)+log(distance to road)+Area | ~year + water | ~Vegetation/ Density | 22 | 1.85 | 0.04 |

consistent differences between areas east and west of the river that had the same legal protection. All species occurred at lower densities in the GMA, ranging from 4.5-fold to 13.2-fold lower than densities within SLNP, with no overlap in 95% CIs.

**Table 4. Coefficient estimates from each species' top group density model.** Covariates included in model averaging for zebra and warthog but not in the top model are also indicated (+).

| $\lambda$—Covariates | Impala β Estimate (SE) | Puku β Estimate (SE) | Zebra β Estimate (SE) | Warthog β Estimate (SE) |
|---|---|---|---|---|
| Intercept | -3.53 (0.98)* | -0.43 (0.45) | 0.43 (0.99) | -1.53 (1) |
| B-Edge Density | 2.36 (0.55)* | 1.71 (1.19) | - | - |
| B-Edge Density$^2$ | 0.70 (0.46) | 0.82 (0.8) | - | - |
| B-Log(Edge Density) | - | - | -0.49 (0.31) | - |
| B-Log(% CS) | - | - | - | -0.46 (1.17) |
| B-Log(% CS)$^2$ | - | - | - | -2.57 (1.02)* |
| B-% CW | - | - | - | 1.31 (2.11) |
| B-% CW$^2$ | - | - | - | -2.55 (1.25)* |
| B-log(% CW) | - | -0.08 (0.02)* | -0.07 (0.02)* | - |
| B-% OW | - | - | - | + |
| B-% OW$^2$ | - | - | - | + |
| B-log(% OW) | - | - | 0.07 (0.15) | - |
| B-% OG | - | - | - | 0.82 (1.67) |
| B-% OG$^2$ | - | - | - | 3.92 (1.21)* |
| B-log(% OG) | - | -0.01 (0.06) | - | - |
| T-Lion UD | - | -0.62 (2.2) | - | - |
| T-Lion UD$^2$ | - | -3.96 (1.29)* | - | - |
| T-log(Lion UD) | 1.34 (0.25)* | - | - | 0.46 (0.22)* |
| A-Log(Distance to Roads) | -0.06 (0.02)* | -2.92 (1.17)* | + | 0.08 (0.04) |
| A-Log(Distance to Roads)$^2$ | - | -7.73 (1.64)* | - | - |
| A-Side: West | 0.60 (0.25)* | -1.67 (0.42)* | 0.71 (0.37) | - |
| A-Area: Park | 0.51 (0.28) | 2.34 (0.48)* | 1.94 (0.69)* | 1.56 (0.47)* |
| AB-Distance to Luangwa | -1.68 (1.20)* | -11.96 (3.09)* | - | - |
| AB-Distance to Luangwa$^2$ | 3.90 (0.78)* | 8.22 (1.8)* | - | - |
| AB-Distance to Seasonal Stream | -0.01 (0.73) | 1.04 (1.34) | - | - |
| AB-Distance to Seasonal Stream^2 | -0.23 (0.61) | 2.29 (1.03)* | - | - |
| $\varphi$—Covariates | β Estimate (SE) | β Estimate (SE) | β Estimate (SE) | β Estimate (SE) |
| Intercept | -1.67 (0.30)* | -1.93 (0.38)* | -3.48 (0.49)* | -1.77 (0.44)* |
| B-Lagoon:Present | 0.26 (0.12)* | 0.34 (0.2) | 0.84 (0.28)* | - |
| B-Grass Height: Short | 0.15 (0.12) | 0.44 (0.15)* | - | - |
| B-Grass Height: Long | -0.05 (0.19) | -0.58 (0.61) | - | - |
| B-Grass Color: Green | - | 0.76 (0.14)* | 0.38 (0.21) | - |
| AB-Water: Presence | 0.47 (0.14)* | - | - | 1.29 (0.27)* |
| AB-Year:2013 | - | - | 0.34 (0.29) | 0.14 (0.26) |
| AB-Year:2014 | - | - | 0.47 (0.29) | -0.26 (0.27) |
| AB-Year:2015 | - | - | 0.97 (0.32)* | -0.96 (0.34)* |
| AB-Season: Mid | 0.05 (0.12) | - | 0.02 (0.27) | - |
| AB-Season: Late | 0.31(0.11)* | - | 0.47 (0.23)* | - |
| AB-Burn: Presence | -0.05 (0.15) | -0.49 (0.29) | - | - |
| $p$—Covariates | β Estimate (SE) | β Estimate (SE) | β Estimate (SE) | β Estimate (SE) |
| Intercept | -3.14 (0.12)* | -2.26 (0.18)* | -2.78 (0.3)* | -3.03 (0.42)* |
| B-Closed Woodland | 0.02 (0.17) | - | 0.72 (0.35)* | -0.24 (0.35) |
| B-Open Scrubland | 0.43 (0.11)* | - | 0.21 (0.32) | 0.2 (0.31) |
| B-Open Woodland | 0.83 (0.11)* | - | 1.1 (0.29)* | 0.58 (0.3) |
| B-Open Grassland Flooded | 0.98 (0.13)* | - | 2.18 (0.93)* | 1.27 (0.43)* |

(*Continued*)

**Table 4.**  (Continued)

|  | Impala | Puku | Zebra | Warthog |
|---|---|---|---|---|
| B-Open Grassland Not Flooded | 1.06 (0.19)* | - | 0.94 (0.5) | 0.66 (0.36) |
| B-Woodland | - | 0.67 (0.17)* | - | - |
| B-Grassland | - | 0.55 (0.14)* | - | - |
| A-Seasonal Track | - | -0.25 (0.18) | -0.82 (0.33)* | - |
| A-Permanent Track | - | -0.33 (0.14)* | -0.46 (0.23)* | - |
| Hazard Detection Function | 0.86 (0.08)* | 0.85 (0.13)* | 0.83 (0.23)* | 0.74 (0.26)* |

Coefficient estimates with p-values<0.05 are indicated (*).

**3.2.2 Temporal effects.**   Density estimates varied among years, but annual differences from regional mean densities were species-specific, with no consistent pattern for across species (Fig 2). Impala density did not detectably change across the study period in any section of the protection gradient. Puku density increased from 2012 to 2013 in western parklands but did not differ from 2014 and 2015 estimates or the overall mean density for that region. Zebra

**Table 5.  Log-link coefficient estimates and standard errors from final herd size models.** Change values are the multiplicative changes of group size based on the associated coefficient estimate, derived using the *predict()* function. Covariates not included in the final models are indicated as either dropped during model refinement (+) or excluded from model refinement due to high correlation or imbalanced sampling (-).

|  | Impala | | Puku | | Zebra | | Warthog | |
|---|---|---|---|---|---|---|---|---|
| Covariate | Estimate (SE) | Change | Estimate (SE) | Change | Estimate (SE) | Change | Estimate (SE) | Change |
| Intercept | 0.612 (0.288)* |  | -0.252 (0.570) |  | 1.977* (0.107) |  | 1.294 (0.103)* |  |
| Mixed Species Herd | 0.563 (0.091)* | 1.72 | 0.617 (0.172)* | 1.27 | 0.228* (0.081) | 1.25 | + | + |
| B-Edge Density | 0.037 (0.005)* | 1.03 | + | + | -0.024* (0.004) | 0.98 | + | + |
| B-Lagoon:Present | + | + | 1.588 (0.378)* | 2.47 | + | + | + | + |
| B-Log(% CS) | 0.028 (0.009)* | 1.02 | + | + | + | + | + | + |
| B-% CW | - | - | - | - | - | - | - | - |
| B-% OW | 0.004 (0.002)* | 1.00 | - | - | + | + | + | + |
| B-% OG | - | - | 0.008 (0.003)* | 1.00 | - | - | + | + |
| B-Grass Height: Short | 0.194 (0.081)* | 1.19 | + | + | 0.360* (0.100) | 1.42 | + | + |
| B-Grass Height: Long | -1.949 (0.478)* | 0.38 | + | + | 0.230 (0.132) | 1.25 | + | + |
| B-Grass Color: Green | -0.562 (0.103)* | 0.58 | -0.357 (0.173)* | 0.88 | + | - | -0.469 (0.117)* | 0.71 |
| T-Lion UD | - | - | - | - | + | + | - | - |
| A-Log Distance to Roads | + | + | + | + | + | + | + | + |
| A-Side: West | -0.275 (0.100)* | 0.79 | -0.807 (0.180)* | 0.84 | - | - | + | + |
| A-Area: Park | - | - | - | - | + | + | - | - |
| AB-Distance to Luangwa | 0.126 (0.020)* | 1.12 | -0.169 (0.087) | 0.96 | - | - | - | - |
| AB-Distance to Seasonal Stream | 0.067 (0.017)* | 1.06 | - | - | + | + | -0.097 (0.029)* | 0.942 |
| AB-Water: Presence | 0.786 (0.100)* | 1.77 | - | - | + | + | + | + |
| AB-Year:2013 | 0.517 (0.125)* | 1.64 | 0.844 (0.290)* | 1.43 | + | + | + | + |
| AB-Year:2014 | 0.339 (0.111)* | 1.38 | 0.293 (0.305) | 1.10 | + | + | + | + |
| AB-Year:2015 | 0.268 (0.132)* | 1.29 | 0.661 (0.324)* | 1.30 | + | + | + | + |
| AB-Season: Mid | 0.319 (0.104)* | 1.35 | 0.244 (0.184) | 1.08 | -0.209 (0.107) | 0.82 | + | + |
| AB-Season: Late | -0.086 (0.103) | 0.93 | -0.401 (0.248) | 0.91 | -0.210* (0.086) | 0.81 | + | + |
| AB-Burn: Presence | - | - | - | - | 0.303* (0.087) | 1.35 | + | + |

* Coefficient estimates with p-values<0.05.

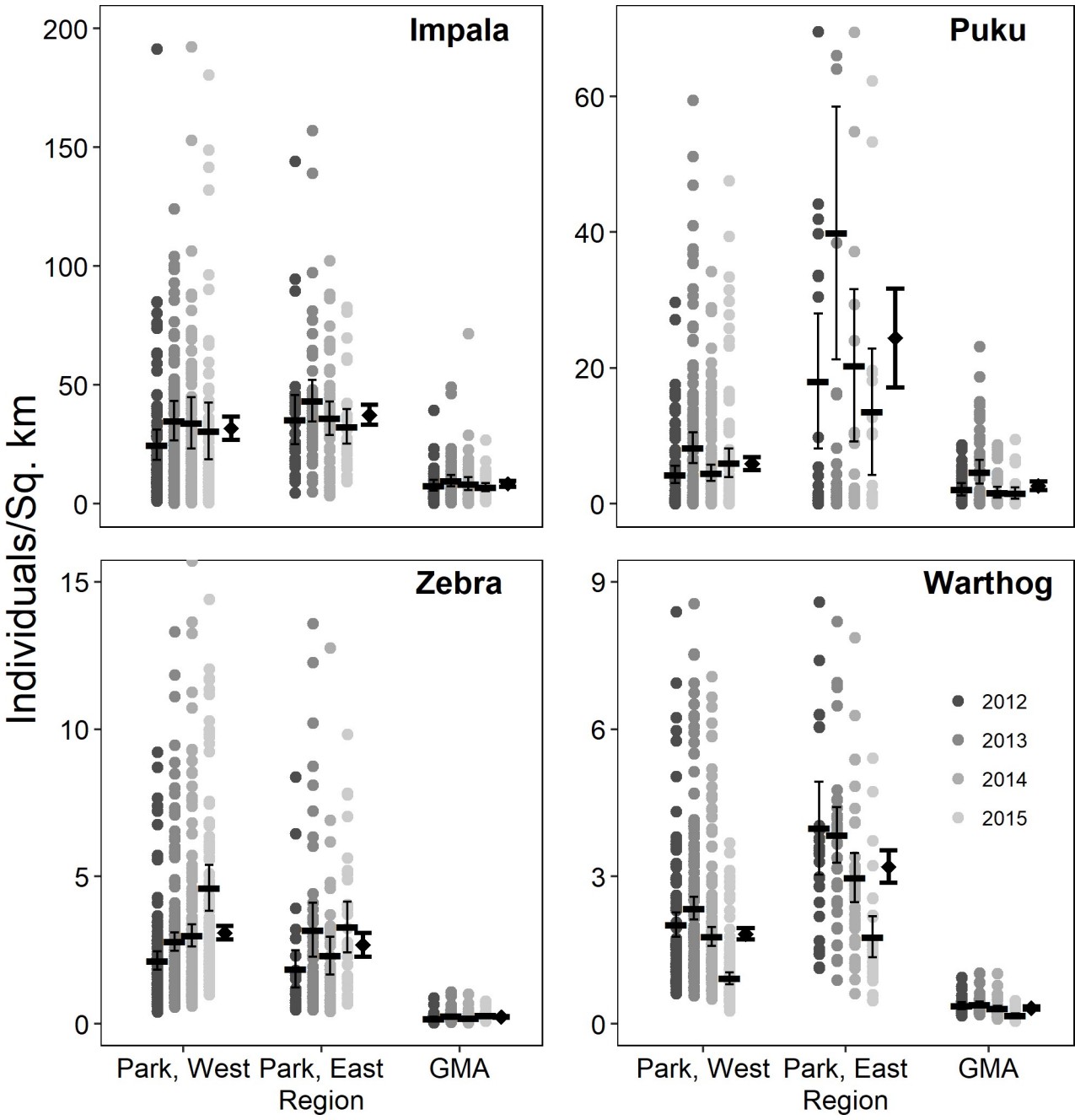

**Fig 2. Population density estimates across the South Luangwa protection gradient.** Regions are arranged left to right in level of protection, with west and east denoting side of the Luangwa River. Points indicate segment-specific density estimates across the four-year study, overlaid with average annual densities and 95% CIs. Overall regional average densities (diamonds) and 95% CIs are displayed in bold alongside the annual averages. Y-axes were truncated to clearly display variation between annual averages, and thus omit extreme segment estimates from being displayed (13 impala, 16 puku, 4 Zebra, and 3 warthog estimates out of 890 total estimates for each species).

densities did not change in GMA and eastern parklands but increased in western parklands. Finally, we detected a decline in warthog densities across all regions in the protection gradient.

**3.2.3 Predicted population density across the SLPG and the study period.** After controlling for all covariates, we found that the observed differences in population density just

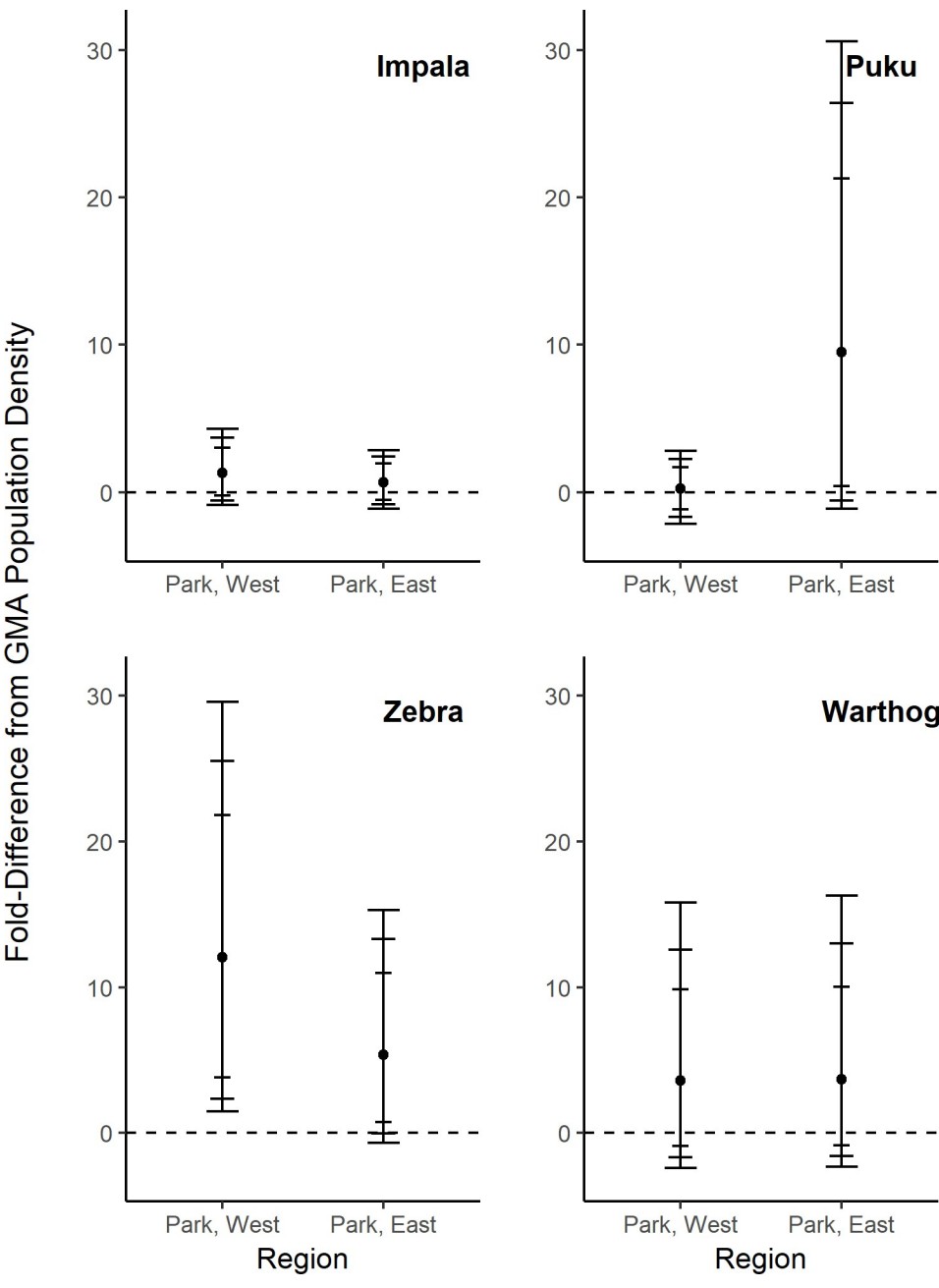

**Fig 3. Differences in population density estimates between protected park lands and GMA population density estimates (dotted line), with other covariates held constant.** Error bars indicate 80%, 90%, and 95% confidence limits.

described were not always detectably related to the level of protection (Fig 3) or year (Fig 4). Across the SLPG, predicted mean species densities were consistently higher in protected areas, but some of these estimated differences had low precision (Fig 3). For impala, populations in protected parklands east and west of the Luangwa River were estimated to have 1.3- and 0.66-fold greater densities compared to GMA densities, but all CIs showed that predicted mean density overlapped with GMA estimates. Puku densities were 9.5-fold greater in the eastern parklands (with >80% confidence that a difference existed due to protection), while

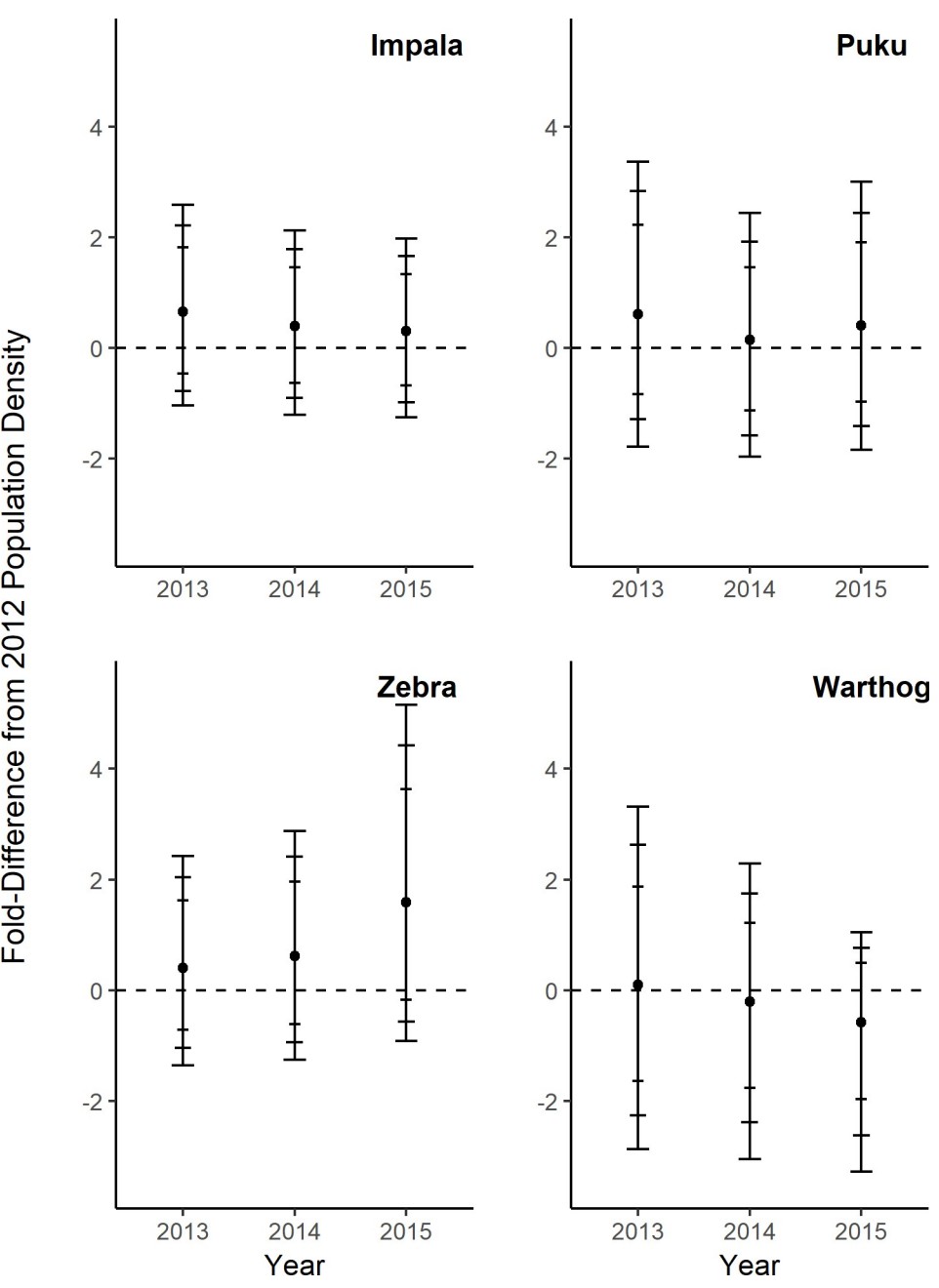

**Fig 4. Differences in population density estimates through the study period and 2012 density estimates (dotted line), with other covariates held constant.** Error bars indicate 80%, 90%, and 95% confidence limits.

densities in western parklands were not detectably different from the GMA. Predicted zebra densities were 5-fold and 12-fold greater in the eastern parklands and western parklands, respectively, with high confidence that a protection effect existed for both comparisons. Predicted warthog densities were higher in protected regions compared to the GMA, but CIs overlapped predicted GMA densities. Throughout the study period we could not detect any density trends for any species after controlling for other covariates (Fig 4), including during the trophy hunting moratorium (2013–2014).

In summary: (1) Ungulate densities were consistently higher in the better protected SLNP. (2) There was evidence that higher protection status yields higher density, after controlling for ecological and abiotic differences between NP and GMA, but this evidence was mixed and sometimes not strong. (3) After controlling for other effects, there was no consistent pattern of change in ungulate densities over the four years of observation.

## 4. Discussion

Our study of the SLPG supports previous findings that lands with better protection generally hold higher densities of herbivores [14–17], supporting continued protection and monitoring in the SLPG. In particular, the segment of SLNP east of the Luangwa River still supports densities of ungulates comparable to or greater than in SLNP west of the Luangwa River, despite having no physical boundary to prevent human incursions and a heavily-used public road passing through it. We estimated the lowest densities for all focal species in the SLPG within the GMA, which supports conclusions from previous studies in the SLPG that suggested depleted herbivore populations within Lupande GMA [23,26]. Given that our GMA transects surveyed areas within 6 km of the Luangwa River, where wildlife densities are thought to be highest and away from dense human settlements, the densities we estimated within this buffer area are likely high compared to the rest of the GMA. These inferences are restricted to dry season conditions, as we were not able to survey transects in wet season conditions.

However, after controlling for a suite of bottom-up, top-down, anthropogenic, and abiotic covariates, there is no clear evidence that these dynamics were driven by protection status or year alone. The non-anthropogenic variables included in our analysis incorporate ecosystem alteration by humans, and thus our interpretation of the role of the SLPG regions and time are proxies for anthropogenic mortality and associated risk effects [48]. Isolating these forces from ecological variables is not entirely clear; for example, poaching efforts are non-random and are correlated with both anthropogenic and ecological variables [27]. We did not allow trends to vary across the SLPG, so any differences in trend between regions for a species are due to covariates other than year (Fig 2), supported by lacking trends when all covariates other than year were held constant. With these considerations, our findings indicate that an array of ecological and anthropogenic variables influence herbivores in protected area networks characterized by national parks (or other strictly protected areas) and adjoining buffer zones.

If we had not applied our rigorous modeling of both group density and size, and instead followed previous approaches (see Section 1), our inferences would have been confined to the variables of interest (protection and year) and would not acknowledge the effects of other important variables. With our approach, ecological variables could play an important role in our modeling process, highlighting the importance of considering ecological conditions in the SLPG along with the protections provided by SLNP. Herbivore density can vary across gradients of protection, but our findings indicate that the protection status of an area is generally insufficient to capture the complexities of factors influencing herbivores across this protection gradient.

### 4.1 Differences in density across the South Luangwa Protection Gradient

Illegal wire snare poaching has been a well-recognized threat to wildlife communities in the Luangwa Valley [30] and has been identified as a major threat to the persistence of large carnivores [26]. In this study, GMA transects represented areas that were prone to high risk for snare occurrence relative to risk in SLNP due to proximity to human activity and greater law enforcement efforts within SLNP [27]. Therefore, we would expect reduced herbivore densities in GMA relative to SLNP regions if bushmeat poaching was impacting herbivore densities.

After controlling for all other variables, our density estimates were higher in PA regions of the SLPG, but only zebra densities reflected this expectation with 95% confidence (puku reflected this expectation with 80% confidence; Fig 3). The detected role of the SLPG in zebra density dynamics indicates evidence of increased impacts of bushmeat poaching and other human activities, particularly as larger ungulate species are vulnerable to overharvest [3]. While we cannot identify bushmeat poaching as the sole driver behind the role of the SLPG in zebra density dynamics (survival data are unavailable), other sources of mortality are unlikely drivers in this study. Trophy hunting occurred with limited quotas and small harvest rates relative to our population density estimates, and lion utilization was accounted for in our modeling of group density and size (Table 4). Therefore, the predictable impact of wire-snare poaching on zebra (and likely puku) is a signal for concern but demonstrates that the protection gradient is providing protection from human exploitation.

Despite concerns of increased poaching in the absence of wildlife-based economies during the trophy hunting moratorium in GMAs, there is no evidence of any coinciding large-scale herbivore decline in our study area. Anti-poaching efforts in the study area were conducted jointly by the Department of National Parks and Wildlife and Conservation South Luangwa during this period, and the level of anti-poaching effort and support increased during the moratorium [29,49–51]. We acknowledge that this is not necessarily the case in all concessions but does indicate the potential for greater anti-poaching investment by operators in these areas. We did not test for interactions between year and region of the protection gradient, so our estimates of population trends could hide local declines in the GMA. However, these results indicate no evidence that the 2013–2014 trophy moratorium was detrimental to the species in this study.

Estimated densities across the SLPG and throughout the study period are clearly influenced by ecological variables, indicating that habitat alteration likely plays an important role in decreasing available resources and reducing herbivore densities in the GMA. Colleagues [19] documented rapid rates of human encroachment in Lupande and other GMAs around key Zambian protected areas. Our GMA transects did not pass directly through any majorly altered areas (e,g, agriculture or villages), but such areas were present across the GMA region of the SLPG. In addition to human encroachment, altered composition of the wildlife community in the GMA, particularly the poaching of elephants and rhinos (extirpated by 1995) [52], likely influences vegetation structure in this region [53]. The role of habitat conversion and changing wildlife community composition should be further investigated in the SLPG, alongside ongoing attention to combat illegal bushmeat poaching.

## 4.2 The future role of ground transects in the South Luangwa Protection Gradient

Future monitoring is critical to track population densities for these and other herbivore species, particularly in the face of a growing human population. While multiple approaches are implemented to achieve this monitoring, we advocate continued monitoring by stratified, ground-based transects. Aerial survey data collected by the Zambia Department of National Parks and Wildlife supports the notion that the SLPG supports notably high densities for all four species [54], and therefore remains an important area to protect and monitor. While ground-based distance sampling surveys cannot match the spatial scale achievable by aircraft, ground-based surveys provide more accurate density estimates to identify population trends and aid in our understanding of the function of protected areas. The difference in costs between the two methods is well-documented [55], and population monitoring could be supplemented with data collected by law enforcement patrols [56] and citizen science initiatives

[57] to offset costs and to achieve a large study area. Ground-based distance sampling methods should be integrated into long-term monitoring of the SLPG and in other protected areas, focused on critical areas for ungulate populations and accounting for the dynamics of ecological covariates in both group density and size, as demonstrated in this study.

## Supporting information

**S1 Appendix. Detection and covariate data from line transect surveys.**
(XLSX)

**S1 Fig. Histogram of ungulate group detections.** Distributions of detections of the four study ungulate species from driven line transects. Detection data are truncated to exclude outlier detections (400m for puku, 300m for impala, warthog, and zebra).
(TIF)

## Acknowledgments

We thank the Zambia Department of National Parks and Wildlife for permission and collaboration with this study. Many thanks to G. Banda, T. Banda, B. Beza, R. Kabungo, A. Mwanza, A. Rosenblatt, E. Sadowski, and M. Sichande for their assistance with data collection.

## Author Contributions

**Conceptualization:** Scott Creel, Paul Schuette, Matthew S. Becker.

**Data curation:** Elias Rosenblatt, Paul Schuette, Egil Dröge.

**Formal analysis:** Elias Rosenblatt, Scott Creel, Paul Schuette, David Christianson, Egil Dröge.

**Funding acquisition:** Scott Creel, Matthew S. Becker.

**Investigation:** Elias Rosenblatt, Scott Creel, Paul Schuette, Matthew S. Becker, Egil Dröge, Thandiwe Mweetwa, Henry Mwape, Johnathan Merkle, Jassiel M'soka, Twakundine Simpamba.

**Methodology:** Scott Creel, Paul Schuette, Matthew S. Becker, David Christianson.

**Project administration:** Scott Creel, Paul Schuette.

**Software:** Elias Rosenblatt.

**Supervision:** Scott Creel, Paul Schuette, Matthew S. Becker, David Christianson.

**Visualization:** Elias Rosenblatt.

**Writing – original draft:** Elias Rosenblatt, Scott Creel, Matthew S. Becker.

**Writing – review & editing:** Elias Rosenblatt, Scott Creel, Paul Schuette, Matthew S. Becker, David Christianson, Egil Dröge, Thandiwe Mweetwa, Henry Mwape, Johnathan Merkle, Jassiel M'soka, Jones Masonde, Twakundine Simpamba.

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
