## [Decision Letter · Decision Letter 0]

14 Aug 2019

PONE-D-19-19082

Do protection gradients explain patterns in herbivore densities? An example with ungulates in Zambia’s Luangwa Valley

PLOS ONE

Dear Dr. Rosenblatt,

Thank you for submitting your manuscript to PLOS ONE. After careful consideration, we feel that it has merit but does not fully meet PLOS ONE’s publication criteria as it currently stands. Therefore, we invite you to submit a revised version of the manuscript that addresses the points raised during the review process.

Both reviewers and I were impressed by the quality of writing and the details of the statistical analysis. The first reviewer suggested providing histograms of the humped detection curves to show what you mean by humped and there were a number of minor comments by reviewer 2 (some in attachment). Please consider these comments and revise as you see fit.

We would appreciate receiving your revised manuscript by September 30, 2019. To enhance the reproducibility of your results, we recommend that if applicable you deposit your laboratory protocols in protocols.io, where a protocol can be assigned its own identifier (DOI) such that it can be cited independently in the future. For instructions see: http://journals.plos.org/plosone/s/submission-guidelines#loc-laboratory-protocols

We look forward to receiving your revised manuscript.

Kind regards,

Floyd W Weckerly

Academic Editor

PLOS ONE

Journal Requirements:

2. We note that Figure 1 in your submission contain [map/satellite] images which may be copyrighted. All PLOS content is published under the Creative Commons Attribution License (CC BY 4.0), which means that the manuscript, images, and Supporting Information files will be freely available online, and any third party is permitted to access, download, copy, distribute, and use these materials in any way, even commercially, with proper attribution. For these reasons, we cannot publish previously copyrighted maps or satellite images created using proprietary data, such as Google software (Google Maps, Street View, and Earth). For more information, see our copyright guidelines: http://journals.plos.org/plosone/s/licenses-and-copyright.

4.  Thank you for stating the following in the Financial Disclosure section:

This research was funded by: 1. WorldWide Fund for Nature–Netherlands, https://www.wnf.nl/, Grantees: Matt Becker; 2. National Science Foundation Animal Behavior Program under IOS-1145749, Website: https://www.nsf.gov/funding/aboutfunding.jsp, Grantees: Scott Creel, Matt Becker, Dave Christianson, Paul Schuette; 3. National Geographic’s Big Cats Initiative, Website: https://www.nationalgeographic.org/projects/big-cats-initiative/, Grantees: Matt Becker; 4. Painted Dog Conservation Inc, Website: http://www.painteddogconservation.iinet.net.au/, Grantees: Matt Becker. The funders had no role in study design, data collection and analysis, decision to publish, or preparation of the manuscript.

We note that you received funding from a commercial source: Painted Dog Conservation Inc

Reviewers' comments:

Reviewer's Responses to Questions

**Comments to the Author**

1. Is the manuscript technically sound, and do the data support the conclusions?

Reviewer #1: Yes

Reviewer #2: Partly

2. Has the statistical analysis been performed appropriately and rigorously? 

Reviewer #1: Yes

Reviewer #2: Yes

3. Have the authors made all data underlying the findings in their manuscript fully available?

Reviewer #1: Yes

Reviewer #2: Yes

4. Is the manuscript presented in an intelligible fashion and written in standard English?

Reviewer #1: Yes

Reviewer #2: Yes

5. Review Comments to the Author

Reviewer #1: This paper uses data from 10 distance sampling surveys conducted over 4 years on 15 transects subdivided into 97 segments to examine variation in 4 species of mammal densities across space and time in relation to biotic and abiotic factors. The paper is very well written, the analysis is comprehensive and appropriate, and the conclusions are sound. The paper is very focused on their specific study area, but has implications for other areas. The detail provided regarding the analysis was much appreciated and allows readers to understand their complex analytical design that gave insight into the complex nature of mammal density variation in savanna systems. One aspect that deserves additional attention is the potential for a 'humped' distribution for some species in their detection curves. The authors addressed this issue, but some readers like myself would like to see the histograms of detection distances so we can see for ourselves what those distributions were. Overall, this is an excellent example of a detailed analysis on a rich and difficult to collect data set from an understudied region. This study serves as an example of an excellent ground monitoring system that compliments wide-scale aerial surveys and elucidates the complex ecological actors affecting mammal population density and distribution dynamics in African savannas. Some readers may wish for more context relating this study to others from the region or similar savanna biomes, but I am happy to read papers like this one that have a local focus and are well written examples of an excellent study.

Reviewer #2: I reviewed ‘Do protection gradients explain patterns in herbivore densities? An example with ungulates in Zambia’s Luangwa Valley’ which was submitted to Plos One by Rosenblatt et al. The paper examines temporal and spatial changes in population estimates of 4 species in Africa to determine whether protection (i.e., a ban on hunting) affects animal density. Overall, the paper was very well written and was very statistically thorough. There are a few items that warrant further explanation/clarification.

1. Authors conducted transect based distance sampling via vehicles in 3 regions – the National Park, a subset of the National Park where poaching was thought to be occurring, and a Game Management Area. The animals in the National Park will be accustomed to vehicular traffic due to the frequent photography safaris whereas the animals in the Game Management Area will likely associate vehicles with hunting activities. How would the different behavior among sites influence analysis, particularly the density estimates?

2. Given that density may be a misleading indicator of habitat quality (Van Horne 1983), I am surprised authors did not supplement data with recruitment data. Data such as adult female:offspring ratios would be easy to obtain during surveys and would elucidate factors contributing towards different density estimates.

3. Would it be possible for distribution of the 4 animals to be widespread during the wet season and become concentrated near the river during the dry season when your surveys took place? If animals do become concentrated, would it be possible for the animals that were East of the National Park to cross the river and enter the National Park where human disturbance is presumably lower and potentially inflate density estimates in the National Park? Some supporting information that describes site fidelity (or lack thereof) of the 4 focal species would be helpful.

4. A few more minor comments can be found within the pdf.

6. PLOS authors have the option to publish the peer review history of their article (what does this mean?). If published, this will include your full peer review and any attached files.

Reviewer #1: No

Reviewer #2: No

---

## [Author Response · Author response to Decision Letter 0]

7 Oct 2019

From the document "Response to Reviewers"

Review Comments to the Author

Reviewer 1 – Reviewer 1’s comments were very supportive of both the work and how it is presented in the manuscript. We appreciate their support and feedback.

1. Reviewer 1’s only suggestion for modification was to include the histograms of detection distances for each species. We have included this figure in the supplemental materials.

Reviewer 2 – Reviewer 2 offered three constructive points for clarification, which we address below in italicized font.

1. “Authors conducted transect based distance sampling via vehicles in 3 regions – the National Park, a subset of the National Park where poaching was thought to be occurring, and a Game Management Area. The animals in the National Park will be accustomed to vehicular traffic due to the frequent photography safaris whereas the animals in the Game Management Area will likely associate vehicles with hunting activities. How would the different behavior among sites influence analysis, particularly the density estimates?”

This is a good point, and text was added to the manuscript to address this concern (Line 137-139 under “Minor Comments” - see below). We did not estimate differences in detection across the three regions of the protection gradient because our primary interest was estimating differences in density across those regions, and we cannot estimate the effect of this variable on both detection and density within the same model. However, we believe that differences in response to vehicles are not likely to influence our density estimates for the following three reasons. First, policy for legal trophy hunting harvest does not allow shooting from a vehicle, so hunting pressures directly from vehicles would be unusual (and illegal; though not impossible). Second, illegal bushmeat poaching is almost never conducted from vehicles in this system. As detailed in the manuscript, wire-snaring of wildlife is the primary method of poaching of the four focal ungulate species. This wire snare poaching is done on foot by small groups of poachers, often in discrete locations to avoid law enforcement. 

2. “Given that density may be a misleading indicator of habitat quality (Van Horne 1983), I am surprised authors did not supplement data with recruitment data. Data such as adult female:offspring ratios would be easy to obtain during surveys and would elucidate factors contributing towards different density estimates.”

The intent here is simply to test whether differences in density can be attributed to protection gradients, after controlling for other important factors that may affect density. If we understand correctly, this comment was made in regards to our statement, “Thus, differences in animal density between PAs and buffer zones can exist due to natural ecological differences between locations unrelated to the effectiveness of their protection” (Lines 59-60). We want to clarify that we do not equate “natural ecological differences” to habitat quality, as demonstrated by the comprehensive suite of variables considered in our group density models. We are using these natural ecological differences to explain variations in group density, after accounting for imperfect detection and availability. We agree that analysis of demographic differences between these areas would also be of interest, but beyond the scope of this analysis.

3. “Would it be possible for distribution of the 4 animals to be widespread during the wet season and become concentrated near the river during the dry season when your surveys took place? If animals do become concentrated, would it be possible for the animals that were east of the National Park to cross the river and enter the National Park where human disturbance is presumably lower and potentially inflate density estimates in the National Park? Some supporting information that describes site fidelity (or lack thereof) of the 4 focal species would be helpful.”

As suggested, there is an increased concentration of wildlife along the Luangwa River during the dry season (Lines 114-117). We agree with the reviewer’s point about the importance of site fidelity for our inferences. While we cannot provide peer-reviewed evidence of this site fidelity, the probability of an unobserved seasonal migration of animals from the GMA to the National Park is very low. The activities of our field teams and of photographic safari operators are centered along the Luangwa River, and even in the height of the dry season, there is very little movement of these four species across the Luangwa River, likely due to the predation risk from crocodiles. Additionally, if these species were migrating across the Luangwa River into the National Park, we would expect to detect an effect of the protection gradient rather than the environmental conditions which we identified. Finally, unpublished data from radio-collared impala and puku collected in a different Zambian national park indicates extreme site fidelity, and we expect similar behavior in the South Luangwa Protection Gradient. 

Minor Comments in the Manuscript

Lines 69 - 60: Already addressed this comment in Response to Reviewers (Reviewer 2, Point 2).

Line 99: Added citation for Lewis & Phiri 1998

Line 100: The sentence was not referring to the negative impact of legal trophy hunting, but the negative impact of increased opportunity for poaching activities during the trophy hunting moratorium (lines 86 – 90).

Line 123: Added region sizes in the following descriptions.

Line 130: Added “leased, unfenced concessions” to sentence as suggested.

Line 137-139: Already addressed this comment in Response to Reviewers (Reviewer 2, Point 1). Added “GMA transects were primarily on primitive roads or off-road, to minimize any sensitivity of animals to vehicle noise. As illegal harvests of wildlife are usually done on foot, these species did not appear to be sensitive to our driven transects” (Lines 148-149).

Line 174: The reviewer is correct that human encroachment is a concern. However, none of the variables included as covariates in the group density or size models provides a precise measure of human encroachment. Landsat 7 was the land survey data available at the time of the study. We believe that we are justified in using Landsat 7 as this imagery was sufficient to detect roles of edge density and vegetation class in group density and group size for the focal species. Additionally, we clumped the 10 original vegetation classes recorded in the Landsat 7 imagery into four general classes, and so that this variable is not sensitive to changes in pixel values between the Landsat 7 (2010) and Landsat 8 (2013) imagery data. A downside of this procedure is that it limits the utility of the images to detect encroachment.

Line 201: Most of these locations were from the dry season in each of the five years. Most of these locations were from direct observation, which were largely only possible in the same period of the year that the line transects could be surveyed (May-November).

Line 296: We appreciate the reviewer voicing their confusion with the differing buffer distances. We used the 240m buffer for vegetation classification based on pixel size of the landsat data (30m) which extends beyond the visual range of observers during the surveys. This distance contained most group observations (see newly added histograms of detection distances, S1 Figure). We truncated the observation datasets to 300m and 400m based on observed distributions of detection distances (as is normal practice in the analysis of distance sampling data, to avoid idiosyncratic detection data at distances where observations become sparse). Because they depend on the data, these thresholds were not predetermined. Despite the difference in these distances, we believe the vegetation buffer appropriately describes vegetation cover along transects.

Line 309: These differences in group density across regions of the protection gradient are reported in Table 4 (Group density model coefficients) and are integrated into the population density plots of Figure 2.

Line 431-434: We do not have survival or recruitment for these species (also see reply to comment above), but because we controlled for a suite of factors influencing density and still detected differences in density across regions with differing bushmeat poaching and human activities, we feel that we have appropriately dismissed alternative drivers. We have modified the passage to reflect the reviewer’s comment: “The detected role of the SLPG in zebra density dynamics indicates evidence of increased impacts of bushmeat poaching and other human activities, particularly as larger ungulate species are vulnerable to overharvest [3]. While we cannot identify bushmeat poaching as the sole driver behind the role of the SLPG in zebra density dynamics (survival data are unavailable), other sources of mortality are unlikely drivers in this study.” We then use the following sentences ruling out other sources of mortality to eliminate alternative hypotheses (Lines 434-438).

Figure 1: The lion locations were recorded from both direct observation and gps-equipped radio collars. Well-studied prides and coalitions were monitored throughout the entire 1200 km study area detailed in this manuscript. Lion locations reflect the distribution of lions, not monitoring effort. We have added a sentence on line 201 to clarify this: “… by measuring the utilization of the area by the lion (Panthera leo) population (the most abundant large carnivore within the study area) [20]. Intensive lion population monitoring was ongoing during this study, with prides and coalitions monitored across the 1200 km study area detailed in this manuscript.” We also added the following sentence to the Figure 1 caption: “Lion locations reflect pride and coalition locations centering around perennial water sources during the dry season, despite intensive and constant lion monitoring across the 1200 km study area.” We appreciate the reviewer’s preference for the study area map from Rosenblatt et al. 2016. However, we chose this study area map for its illustration of how each variable of interest varies across the area.

---

## [Editor Report · Decision Letter 1]

15 Oct 2019

Do protection gradients explain patterns in herbivore densities? An example with ungulates in Zambia’s Luangwa Valley

PONE-D-19-19082R1

Dear Dr. Rosenblatt,

We are pleased to inform you that your manuscript has been judged scientifically suitable for publication and will be formally accepted for publication once it complies with all outstanding technical requirements.

I have read the revised draft and you have addressed all reviewer concerns adequately. I see no need to send it out for another round of reviews. Below I make a few edit suggestions.

With kind regards,

Floyd W Weckerly

Academic Editor

PLOS ONE

Additional Editor Comments (optional):

Ln 91: ..unbiased and precise estimates…

Ln 98: ..important for ecosystem health… is a vague catch all applicable to any community level study that addresses environmental issues. In my opinion, the fragment has little scientific merit.

 References: please check, seems scientific names are not italicized. Sometimes all words in titles are capitalized, so on.

---

## [Editor Report · Acceptance letter]

21 Oct 2019

PONE-D-19-19082R1 

Do protection gradients explain patterns in herbivore densities? An example with ungulates in Zambia’s Luangwa Valley 

Dear Dr. Rosenblatt:

I am pleased to inform you that your manuscript has been deemed suitable for publication in PLOS ONE. Congratulations! Your manuscript is now with our production department. 

With kind regards,

on behalf of

Dr. Floyd W Weckerly 

Academic Editor

PLOS ONE